

# Decoding biomaterial-associated molecular patterns (BAMPs): influential players in bone graft-related foreign body reactions

Carel Brigi[1], K.G. Aghila Rani[1], Balachandar Selvakumar[1], Mawieh Hamad[1,2], Ensanya Ali Abou Neel[1,3] and A.R. Samsudin[1,4]

[1] Research Institute for Medical and Health Sciences, University of Sharjah, Sharjah, University City, United Arab Emirates
[2] Department of Medical Laboratory Sciences, University of Sharjah, Sharjah, United Arab Emirates
[3] Department of Preventive and Restorative Dentistry, College of Dental Medicine, University of Sharjah, Sharjah, United Arab Emirates
[4] Oral and Craniofacial Health Sciences Department, College of Dental Medicine, University of Sharjah, Sharjah, United Arab Emirates

## ABSTRACT

Bone grafts frequently induce immune-mediated foreign body reactions (FBR), which hinder their clinical performance and result in failure. Understanding biomaterial-associated molecular patterns (BAMPs), including physicochemical properties of biomaterial, adsorbed serum proteins, and danger signals, is crucial for improving bone graft outcomes. Recent studies have investigated the role of BAMPs in the induction and maintenance of FBR, thereby advancing the understanding of FBR kinetics, triggers, stages, and key contributors. This review outlines the stages of FBR, the components of BAMPs, and their roles in immune activation. It also discusses various bone grafting biomaterials, their physicochemical properties influencing protein adsorption and macrophage modulation, and the key mechanisms of protein adsorption on biomaterial surfaces. Recent advancements in surface modifications and immunomodulatory strategies to mitigate FBR are also discussed. Furthermore, the authors look forward to future studies that will focus on a comprehensive proteomic analysis of adsorbed serum proteins, a crucial component of BAMPs, to identify proteins that promote or limit inflammation. This understanding could facilitate the design of biomaterials that selectively adsorb beneficial proteins, thereby reducing the risk of FBR and enhancing bone regeneration.

## INTRODUCTION

Damaged and deformed bone tissue can result from trauma, infections, tumors, and degenerative diseases. Larger bone defects do not heal spontaneously and require bone grafts to support regeneration. Surgeries to correct such defects have a global prevalence of 2.2 million per year (*Ghelich et al., 2022*). Optimally, bone grafting biomaterials, whether metals, biopolymers, or composites, should not induce significant host inflammatory

Corresponding author
A.R. Samsudin,
drabrani@sharjah.ac.ae

responses, should enhance bone regeneration, and provide sustainable mechanical support (*Daculsi et al., 2013*; *Zhao et al., 2021*). However, clinical experience with various biomaterials suggests they often trigger undesirable host reactions or foreign body reactions (FBRs), potentially rendering them ineffective. Numerous studies and case reports have documented failed bone grafts due to FBR (*Adams, 2022*; *Badiee, Rowland & Sun, 2022*; *Elakkiya, Ramesh & Prabhu, 2017*; *Kaing, Grubor & Chandu, 2011*; *Kamata, Sakamoto & Kishi, 2019*; *Lorenz et al., 2016*; *Nonhoff et al., 2024*).

FBR interferes with wound healing, leading to excessive inflammation, severe pain, tissue destruction, graft isolation, and rejection. Although various immune cells participate in the response to bone graft implantation, macrophages play a pivotal role in FBR by phagocytosing foreign materials and recruiting other immune cells to the implantation site (*Lee et al., 2019*; *Li et al., 2023a*; *Ping et al., 2021*; *Sheikh et al., 2015a*; *Wynn & Barron, 2010*).

Recent evidence indicates that macrophages are modulated into pro-inflammatory or anti-inflammatory subsets based on the proteins adsorbed onto biomaterial surfaces (*Blackman et al., 2024*; *Visalakshan et al., 2019*; *Wang et al., 2022*). Proteomic profiling indicates that the surface properties of biomaterials influence protein adsorption, thereby shaping immune cell responses (*Acharya et al., 2010*; *Acharya et al., 2011*; *Blackman et al., 2024*; *Swartzlander et al., 2015*; *Wang et al., 2022*; *Wei et al., 2021*). Consequently, the physicochemical properties of bone grafts, such as surface roughness, wettability, charge, and porosity, are key determinants of biomaterial success or failure.

Biomaterial-associated molecular patterns (BAMPs) encompass (i) the physicochemical properties of biomaterials, (ii) the adsorbed serum proteins, and (iii) the danger signals released by injured cells during bone grafting procedures. The concept of BAMPs suggests that the physicochemical properties of biomaterials regulate the adsorption of serum proteins, which in turn influences the immune cell response to biomaterials (*Abdallah et al., 2017*; *Wang et al., 2022*).

While FBR has been widely researched, this review distinctively highlights BAMPs as a fundamental concept for comprehending immune responses to bone graft materials. It explores how protein adsorption, the physicochemical characteristics of biomaterials, and danger signals influence macrophage phenotypes, significantly influencing the integration and success of bone grafts. Additionally, it closely analyzes how serum proteins adsorbed on biomaterial surfaces influence immune cell activation, with a particular emphasis on macrophage polarization.

While many studies have examined bone grafting biomaterials and immune interactions, few have specifically highlighted the role of BAMPs in FBR. Most reviews have focused on how biomaterial surface properties affect macrophages, overlooking the critical role of protein adsorption mechanisms (*Li et al., 2023a*; *Sheikh et al., 2015a*; *Sun et al., 2024*). To further elaborate, existing literature emphasizes that the surface characteristics of bone grafts, such as topography, wettability, charge, and composition, influence macrophage phenotypes. However, these studies often overlook the profiling of proteins adsorbed on various bone graft surfaces and the resulting immunological responses. Therefore, this

article addresses this knowledge gap by exploring the interplay between biomaterial surface properties, protein adsorption, and immune responses through the concept of BAMPs.

## SURVEY METHODOLOGY

We searched PubMed and ScienceDirect databases for peer-reviewed articles focusing on (a) bone grafts and macrophages, (b) foreign body reaction (FBR) and macrophage modulation, (c) immunomodulation and implanted biomaterials, and (d) protein interactions and biomaterials, covering publications from January 2010 to October 2024.

For bone grafts and macrophages, we used "bone grafting biomaterials" as a basic query and added "macrophages" as a keyword. The search yielded various original research articles and reviews. Of the 6,525 publications retrieved from PubMed for "bone graft biomaterials" with the filter applied for those published in the last 14 years (2010–2024), 946 articles were categorized as reviews, systematic reviews, or meta-analyses. Among these, 14 review articles specifically cited the role of macrophages in bone grafting biomaterials and were chosen for analysis.

For foreign body reaction and macrophage modulation, we used the key search terms "foreign body reaction" AND "macrophage modulation" to gather information on the role of macrophages in FBR. To further narrow the focus on serum proteins regulating macrophages in FBR, we added "protein adsorption" as an additional keyword. The "FBR AND macrophages" query yielded 619 publications, which included 67 meta-analyses and systematic reviews. In writing this review, we concentrated on the role of macrophages in FBR and protein adsorption, identifying seven articles that specifically described the relationship between protein adsorption and macrophage response in FBR.

To assess the immunomodulatory effect on implanted biomaterials, we employed the search terms "immunomodulation" AND "implanted biomaterials", which resulted in 145 publications, including meta-analyses, reviews, and systematic reviews published from 2,010 to 2024. To further investigate the role of BAMPs in regulating FBR, we included "immune cells" as a search term, resulting in 54 articles. We filtered for the most recent studies from 2020 onward, with 27 articles considered for this review. In addition, we searched for "surface properties" AND "macrophage modulation" to assess how physicochemical properties influence macrophage responses, retrieving 78 review articles.

For protein adsorption on biomaterials, we used "protein interactions" AND "biomaterials" as the primary query in ScienceDirect, covering studies from 2010 to 2024. This search yielded 14,865 articles. We refined the results using an advanced search with filters for the title, abstract, and keywords, including "bone regenerative biomaterials", "protein adsorption", and "surface characteristics". This narrowed the selection to 11 relevant research articles, review articles, and book chapters.

The criteria for inclusion in this review encompassed articles published in English and indexed in PubMed or ScienceDirect from January 2010 to October 2024. Both review articles and original research focusing on *in vitro* or *in vivo* studies related to bone graft surface characteristics and macrophage phenotypic differentiation were included. Studies

published before 2010, commentaries, summaries, editorials, and duplicate studies were excluded. Additionally, research on non-bone grafting biomaterials and studies examining immune cells other than macrophages and their roles in immunomodulation were also excluded.

## THE AUDIENCE THIS REVIEW IS INTENDED FOR

The scientific literature review may be especially relevant for bone graft manufacturers, orthopedic surgeons, and dental surgeons. Exploring the relationship between adsorbed host serum proteins and the physicochemical properties of bone grafts in macrophage modulation will deepen the understanding of the FBR process in bone grafts. A more thorough investigation into this concept will help bone graft manufacturers to design grafts that promote the adsorption of FBR-limiting proteins, ultimately improving success rates. Additionally, this review aims to motivate researchers to conduct future studies focused on identifying the adsorbed proteomic profile and its conformational changes on bone graft surfaces.

## BONE GRAFTING BIOMATERIALS

Bone grafts are biomaterials used in dental surgery and orthopedic medicine to replace missing bone due to pathological deterioration, trauma, or accidents. The flowchart in Fig. S1 illustrates the most commonly used biomaterials in bone grafting and regeneration procedures. These biomaterials are classified as osseous (bone or bone-like substances) or non-osseous. The osseous category includes autografts (from the same individual), allografts (from different individuals of the same species), and xenografts (from different species) (*Ferraz, 2023*). Non-osseous biomaterials encompass both metallic and non-metallic substances. Titanium is a widely used metallic biomaterial in orthopedics as well as in oral and maxillofacial surgery. The non-metallic category comprises both inorganic and organic materials. Inorganic bone grafting biomaterials include bioactive glasses and calcium phosphates, such as hydroxyapatite (HA) (*Miron, 2024*; *Wickramasinghe, Dias & Premadasa, 2022*).

Organic materials encompass both synthetic and natural polymers. Natural polymers mainly consist of proteins and polysaccharides, with notable examples including collagen and chitin. These materials are highly biocompatible and suitable for scaffold fabrication because of their similarity to the natural extracellular matrix (ECM). Synthetic polymers such as poly (caprolactone) (PCL), poly (glycolic acid) (PGA), polyether ether ketone (PEEK) and poly (lactic-co-glycolic acid) (PLGA) demonstrate a high bone-inductive potential (*Feng et al., 2018*; *Feng et al., 2023*; *Shuai et al., 2022*; *Shuai et al., 2021*; *Wickramasinghe, Dias & Premadasa, 2022*). The advantages and disadvantages of different bone grafting materials are summarized in Table S1.

The ideal properties of bone grafts include (a) osteoconductivity, which promotes the deposition of new bone matrix; (b) osteoinductivity, which involves the recruitment and differentiation of mesenchymal stem cells into mature osteoblasts to generate bone matrix;
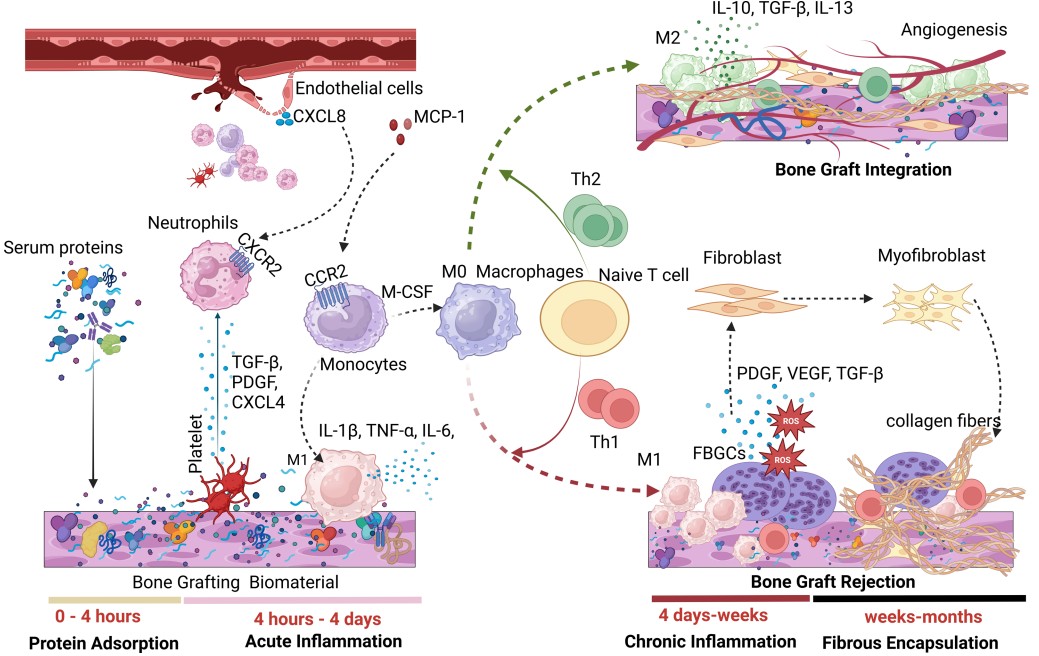

**Figure 1 A tentative rendering of the various phases of FBR.** During the first phase of FBR, serum proteins quickly adsorb onto bone graft surfaces, triggering an acute inflammatory response characterized by neutrophils and M1 macrophages. Persistent M1 macrophages maintain chronic inflammation through Th1 responses, resulting in fibrosis and biomaterial failure. In contrast, the presence of M2 anti-inflammatory macrophages promotes angiogenesis and successful biomaterial integration through Th2 response. (Image created using Biorender.com).

and (c) scaffolding potential, which supports three-dimensional tissue ingrowth (*Miron, 2024*; *Xie et al., 2020*).

## STAGES OF FOREIGN BODY REACTION (FBR)

FBR is an inflammatory and wound-healing response triggered by the implantation of a medical device, prosthesis, or biomaterial (*Albrektsson, Buser & Sennerby, 2012*; *Ivanovski & Mark, 2022*). This process involves a highly orchestrated immune response, characterized by various immune cells and complex biochemical signaling. The FBR follows a sequential progression, beginning with protein adsorption, followed by acute and chronic inflammation, and terminating in biomaterial encapsulation by fibrous tissue (*Zhou & Groth, 2018*). These events ultimately impair the performance and longevity of the biomaterial or prosthesis, often resulting in failure.

A hallmark of FBR is the presence of foreign body giant cells (FBGCs), formed through macrophage fusion. Figure 1 illustrates the phases of FBR during the implantation of biomaterials such as dental bone grafts. FBR can generally result in either foreign body equilibrium and osseointegration at the bone-biomaterial interface or fibrotic encapsulation, leading to implanted bone graft failure (*Trindade et al., 2016*).

## Protein adsorption phase of FBR

FBR begins with forming a provisional matrix as plasma proteins adsorb onto the surface of the implanted biomaterial (*Davenport Huyer et al., 2020*). After implantation, host blood proteins, including albumin, fibronectin, vitronectin, fibrinogen, immunoglobulins, coagulation and complement factors, rapidly adsorb to the biomaterial's surface, primarily within the first four hours. The type and concentration of these adsorbed proteins influence subsequent cellular events and the inflammatory responses of FBR. For instance, fibrinogen adsorption increases macrophage production of the pro-inflammatory cytokine TNF-α, as its P2 domain interacts with integrin αX/β2 on M1 macrophages, triggering a pro-inflammatory response (*Lee et al., 2019*; *Zhou & Groth, 2018*).

In addition to protein deposition, blood-biomaterial interaction activates both the complement and coagulation cascades, forming a provisional matrix rich in fibrin that surrounds the biomaterial. The conversion of fibrinogen to fibrin during clotting generates fibrinopeptides that increase vascular permeability and promote leukocyte chemotaxis (*Binder et al., 2017*). Furthermore, the cleavage of complement factors C3 and C5 releases the anaphylatoxins C3a and C5a, which enhance vascular permeability, chemotaxis, and leukocyte extravasation. Complement activation also stimulates platelet activation and contributes to coagulation through platelet-related coagulation factor IV and the release of clotting factors and activators (*Eriksson et al., 2019*; *Kizhakkedathu & Conway, 2022*).

Protein adsorption on biomaterial surfaces triggers platelet activation, activates the complement system, and promotes coagulation, creating a fibrin-rich matrix around the implant. Furthermore, protein adsorption plays a vital role in modulating macrophage phenotypes at the implantation site.

## Acute inflammation phase of FBR

Platelets within the fibrin mesh of the provisional matrix release cytokines and chemokines, aiding in the recruitment of immune cells. Transforming growth factor-beta (TGF-β), platelet factor 4 (CXCL4), and platelet-derived growth factor (PDGF) from platelets attract neutrophils to implanted bone grafts (*Gleissner et al., 2010*; *Pitchford, Pan & Welch, 2017*). Additionally, CXCL8 released from surrounding endothelial cells interacts with C-X-C chemokine receptor type 2 (CXCR2) on neutrophils, directing them to the bone graft site. Neutrophils are the first immune responders to biomaterial implantation (*Abaricia et al., 2021a*). Following this, circulating monocytes are recruited and differentiate into macrophages, mainly driven by monocyte chemoattractant protein-1 (MCP-1) binding to C–C motif chemokine receptor 2 (CCR2).

During the acute inflammatory stage, macrophages polarize into the M1 phenotype, secreting pro-inflammatory cytokines such as IL-1β, TNF-α, IL-6, IL-12, IL-18, macrophage inflammatory protein 1α/β (MIP-1α/β), and MCP-1 (*McKiel, Woodhouse & Fitzpatrick, 2020*). Unlike short-lived neutrophils, macrophages can persist around the implanted bone graft for several months (*McKiel, Woodhouse & Fitzpatrick, 2020*). This acute inflammatory phase may lead to tissue restoration and osseointegration of bone grafts (*restitutio ad integrum*) or progress into chronic inflammation, resulting in fibrous encapsulation and graft failure (*Ivanovski & Mark, 2022*). The sustained presence of classically activated

macrophages (caMac) or pro-inflammatory macrophages tends to drive the FBR process toward fibrotic encapsulation, while alternatively activated macrophages (aaMac) or anti-inflammatory macrophages promote osseointegration and graft success.

Macrophages also affect the adaptive immune response to bone graft implantation. M1 macrophages, through cytokines such as IL-12, CXCL9, and CXCL10, drive the recruitment and polarization of Th1 cells. In contrast, M2 macrophages release IL-10, CCL17, and CCL22, which promote Th2 responses. An enhanced Th1 response around the bone graft leads to fibrotic encapsulation and graft rejection, while increased Th2 activity supports osseointegration and graft success (*Davenport Huyer et al., 2020*).

## Chronic inflammatory phase of FBR

FBR can progress into a chronic inflammatory stage due to a predominant M1 macrophage population or Th1 response, the release of toxic or degraded biomaterial byproducts, movement of the biomaterial at the implantation site, or inadequate mechanical compliance, including overloading or underloading conditions (*Carnicer-Lombarte et al., 2021*; *Davenport Huyer et al., 2020*). In the later stages of chronic inflammation, macrophages fuse to form FBGCs, which are a hallmark of FBR (*Stewart et al., 2024*). The formation of FBGCs on implanted biomaterials is primarily induced by IL-4 and IL-13 (*Eslami-Kaliji et al., 2023*). *McNally & Anderson (2011)* investigated the effects of adsorbed proteins on polystyrene substrates coated with complement factors C3bi, collagens, fibrinogen, plasma fibronectin, laminin, thrombospondin, vitronectin, and von Willebrand factor to assess monocyte adhesion, macrophage development, and IL-4-induced FBGC formation. While all adsorbed proteins facilitated monocyte adhesion, only vitronectin significantly promoted macrophage growth and FBGC formation. This indicates that surfaces favoring vitronectin adsorption may drive macrophage activation and FBGC formation (*Eslami-Kaliji et al., 2023*; *McNally & Anderson, 2011*; *Sheikh et al., 2015a*). FBGCs contribute to FBR-related fibrosis by releasing reactive oxygen species (ROS) and enzymes that degrade the biomaterial (*Eslami-Kaliji et al., 2023*). Additionally, profibrotic factors, such as PDGF, vascular endothelial growth factor (VEGF), and TGF-β, released from FBGCs, facilitate fibroblast recruitment, further promoting fibrosis (*Zhou & Groth, 2018*).

## Fibrous encapsulation phase of FBR

FBGCs expressing PDGF and TGF-β stimulate fibroblast proliferation, collagen synthesis, and wound healing (*Eslami-Kaliji et al., 2023*; *Zhou & Groth, 2018*). Additionally, FBGCs promote the differentiation of fibroblasts into myofibroblasts by inducing α-smooth muscle actin (α-SMA) expression. During normal wound healing and biomaterial integration (or foreign body equilibrium), myofibroblasts in the surrounding tissue either undergo apoptosis or enter a quiescent state, halting collagen production (*Lebonvallet et al., 2018*). However, in chronic inflammation, myofibroblasts persist and continue to produce excessive collagen fibers, resulting in extensive fibrosis and scarring (*McKiel, Woodhouse & Fitzpatrick, 2020*; *Noskovicova, Hinz & Pakshir, 2021*). This fibrotic response hinders bone graft integration by restricting oxygen and nutrient transport to surrounding tissues, ultimately compromising graft function (*Capuani et al., 2022*).

# FBR AND INFLAMMATORY RESPONSE TO BONE GRAFTING BIOMATERIALS: *IN VITRO* AND *IN VIVO* STUDIES

### *In vitro* studies

THP-1 monocytes exposed to titanium dioxide nanoparticles ($TiO_2$ NPs) and microparticles ($TiO_2$MPs), with sizes <100 nm and <5 $\mu$m, respectively, exhibited an enhanced inflammatory response during *in vitro* analysis. Elevated ROS levels confirmed the uptake of these particles, and the resulting inflammation was compared to controls, and the increase in ROS generation with $TiO_2$NPs was concentration-dependent (*Kheder, Soumya & Samsudin, 2021*). Additionally, research conducted by our group employed human peripheral blood monocyte-derived macrophages (PBMMs) to assess the immunological response of demineralized (DMB) and decellularized (DCC) bovine bone substitutes. The findings indicated that PBMMs treated with DMB demonstrated increased expression of inflammatory cytokine markers IL-1$\beta$ and TNF-$\alpha$, along with proinflammatory cell surface markers CD86 and CD14, while DCC substitutes exhibited immunoregulatory effects on PBMMs (*Rani et al., 2024*). In another *in vitro* study (*Toledano-Serrabona et al., 2022*), researchers utilized titanium metal particles released during implantoplasty of dental implants on macrophage cell cultures (THP-1). The results indicated an increased pro-inflammatory expression of TNF-$\alpha$ and a decreased expression of anti-inflammatory markers TGF-$\beta$ and CD206. These findings suggest that titanium particles play a role in developing bone resorption or peri-implant tissue inflammatory response.

### *In vivo* animal models and human studies

The study by *Ciobanu et al. (2024)* investigated the treatment of critical-sized bone defects (CsBDs) in a rat model using four approaches: untreated defects, defects treated with Bio-Gen®, Bio-Gen® combined with platelet-rich fibrin (PRF), and autologous bone grafts (ABG). The ABG group achieved the most successful healing outcomes. In the Bio-Gen® group, histological analysis revealed the formation of a fibrous callus with numerous capillaries, a giant cell reaction to the bone graft fragments, and sparse lymphocytes. Combining PRF with Bio-Gen® enhanced healing compared to Bio-Gen® alone, with improved tissue regeneration, reduced inflammation, and better vascularization. A study (*Fernandes et al., 2024*) on critical-size calvarial defects in 50 Wistar rats compared to blood (G1), autogenous bone (G2), bioglass (G3), hydroxyapatite (G4), and xenograft (G5) grafts, with or without expanded polytetrafluoroethylene (e-PTFE) barriers. Autogenous bone (G2) demonstrated the best bone formation and resorption outcomes, followed by G4, G5, and G3. Synthetic biomaterials (G3 and G4) yielded comparable results, while G5 resulted in 22% new bone formation after 45 days. Among the synthetic materials, G4 showed a superior degradation profile.

An *in vivo* study on mineralized collagen-polycaprolactone implants in a porcine ramus critical-size defect model found that only 2 out of 22 implants achieved effective bone regeneration, whereas the majority showed limited bone formation and fibrous encapsulation (*Dewey et al., 2021*). Another group (*Tanneberger et al., 2021*) explored

the cellular response to porcine-derived resorbable collagen membranes in Wistar rats over a span of 30 days. The membrane induced mononuclear cell infiltration, forming multinucleated giant cells (MNGCs) by day 15. These cells increased in number and migrated centrally by day 30, expressing CD-68, calcitonin receptor, and MMP-9. The disintegration of the collagen membrane was linked to MNGC activity and significantly increased vascularization compared to the controls. Another group of researchers studied the foreign body response of PCL scaffolds implanted into the dorsal window chamber of 10–12-week-old C57BL/6 mice. Over two to four weeks, they observed the formation of an immature neurovascular network alongside the development of a dense fibrous capsule (*Dondossola et al., 2016*). An *in vivo* study on Beagle dogs found that hydroxyapatite-coated poly-l-lactic acid (PLLA) screws exhibited superior biocompatibility, reduced inflammation, and improved bone integration compared to uncoated PLLA screws, which resulted in significant foreign body reactions characterized by the formation of fibrous tissue and infiltration by histiocytes (*Akagi et al., 2013*).

Histological analysis of 14 tissue samples from patients who underwent sinus augmentation before tooth implantation revealed mild inflammatory responses, including increased immune cells and blood vessels around both xenogeneic (Bio-Oss®) and synthetic (NanoBone®) bone substitutes. Multinucleated giant cells were observed more frequently on the synthetic material, indicating a stronger immune response compared to the xenogeneic substitute (*Barbeck et al., 2017*). A randomized clinical trial (*Koo et al., 2020*) examined bone formation after grafting periodontally damaged extraction sockets using deproteinized bovine bone mineral (DBBM) or deproteinized porcine bone mineral (DPBM) with collagen membrane coverage. A total of 100 patients participated, and 81 biopsy samples (42 from the DBBM group and 39 from the DPBM group) were included in the final analysis. Both groups showed comparable histologic bone formation, although some specimens from both groups exhibited fibrous encapsulation of biomaterial particles in the coronal region.

## THE INTERPLAY BETWEEN BAMPS AND THE FBR

Several recent reports have elaborated on how the physicochemical properties of biomaterials help determine the adsorbed proteomic profile and subsequent cellular interactions (*Abdallah et al., 2017*; *Blackman et al., 2024*; *Wang et al., 2022*). BAMPs are molecular components or characteristics found on biomaterials that can interact with the body's immune system and trigger an inflammatory response. First described by Babensee, BAMPs consist of three main components: (i) adsorbed proteins, (ii) danger signals including DNA, RNA, high-mobility group box-1 (HMGB1), and heat shock proteins (HSPs), and (iii) the physicochemical properties of the biomaterial (*Wang et al., 2022*). BAMPs are analogous to inflammatory stimuli such as pathogen-associated molecular patterns (PAMPs) and damage-associated molecular patterns (DAMPs). BAMPs have been shown to play an important role in FBR, as the initial stage of FBR involves protein adsorption on the surface of the bone graft, which depends on the surface properties of the grafts. Introducing the concept of BAMPs has enabled the examination of how adsorbed

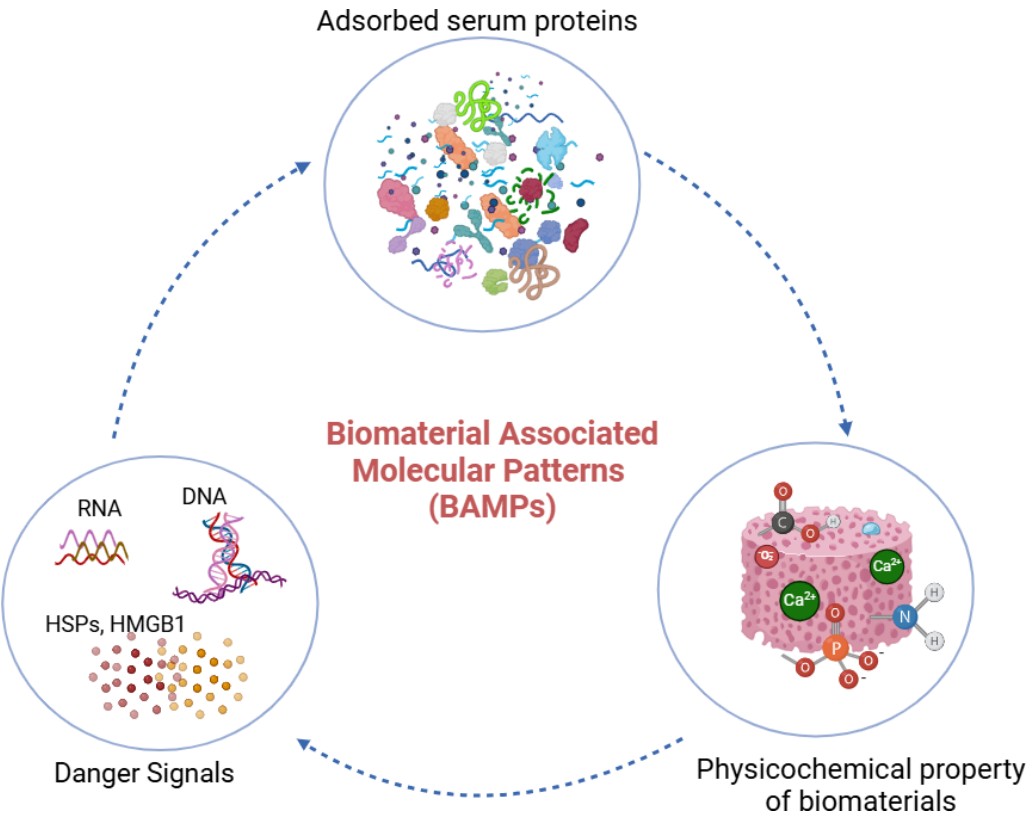

Adsorbed serum proteins

**Biomaterial Associated Molecular Patterns (BAMPs)**

RNA    DNA

HSPs, HMGB1

Danger Signals

Physicochemical property of biomaterials

**Figure 2  BAMPs in bone grafting biomaterial.** Adsorbed serum proteins, danger signals, and the physicochemical properties of the biomaterial comprise the components of BAMPs. The properties of the biomaterial's surface dictate the adsorbed proteome profile and subsequent immune cell interactions. Protein-protein interactions also occur during protein adsorption onto biomaterials. (Image created using Biorender.com).

proteins affect the activity of key immune cells like macrophages in the context of the FBR. The components of BAMPs are detailed in Fig. 2. The concept of BAMPs indicates that the surface properties of biomaterials affect the adsorbed proteomic profiles and the resulting cellular interactions (Abdallah et al., 2017).

## Biomaterial proteins adsorption and their fundamental mechanism

A dynamic layer of adsorbed proteins forms when bone grafts are placed into the host because their surface characteristics allow both the adsorption and desorption of blood proteins. As a component of BAMPs, the adsorbed protein consists of either autologous proteins (found in blood or extracellular fluid) or allogenic proteins (derived from materials). Autologous host serum proteins begin to adsorb at the millisecond level, and a protein layer will have formed by the time immune cells are attracted to the biomaterial implantation site (Eslami-Kaliji et al., 2020; Wang et al., 2022).

It is anticipated that the "big twelve" blood-derived host proteins, which are typically present in human plasma at concentrations of one mg/ml or higher, will compete for

the first interaction on the biomaterial's surface. These proteins include albumin, α-macroglobulin, haptoglobin, low-density lipoprotein (LDL), high-density lipoprotein (HDL), fibrinogen, transferrin, α-antitrypsin, complement factor C3, IgG, IgA, and IgM (*McKiel, Woodhouse & Fitzpatrick, 2020*). High molecular weight proteins and those present in higher concentrations, such as albumin, immunoglobulins, fibrinogen, factor XII (Hageman factor), and high molecular weight kininogen (HMWK), are the first to arrive and initially adhere to the surface of bone grafts. Eventually, these proteins will be replaced by medium- and low-molecular-weight proteins with strong surface affinity. This phenomenon of protein adsorption and desorption on the biomaterial surface is called the Vroman effect (*Wei et al., 2021*).

A variety of factors influence the adsorption of proteins on bone graft surfaces. These include the concentrations and chemical properties of blood proteins, their protein-protein interactions, and the physicochemical characteristics of biomaterials (such as surface area, hydrophobicity, charge, roughness, thickness, porosity, and chemical composition) (*Adams et al., 2019*; *Barberi & Spriano, 2021*; *Ping et al., 2021*; *Stanciu & Diaz-Amaya, 2021*). The rate of protein adsorption is directly related to protein concentration and inversely related to protein molecular weight. Van der Waals forces, electrostatic interactions, hydrophobic interactions, and hydrogen bonds play a role in protein adsorption. Some proteins are reversibly adsorbed on the surfaces of biomaterials and tend to desorb over time. However, others are unlikely to desorb as they bond permanently to the surface. The interactions between proteins and the biomaterial's surface and between proteins determine the final profile of the adsorbed protein layer (*Stanciu & Diaz-Amaya, 2021*; *Talha et al., 2019*).

In an actual scenario, protein adsorption occurs when multiple serum proteins interact with one another simultaneously, as seen in the case of blood plasma. Protein-protein interactions influence protein adsorption on the surface of biomaterials (*Zheng, Kapp & Boccaccini, 2019*). For instance, these interactions between proteins can either cooperative or competitive protein adsorption on the surface of biomaterials. Adsorption, where deposited proteins influence the adsorption of "new" proteins (those that are not adsorbed), is known as cooperative adsorption (*Liu, 2015*). In contrast, competitive protein adsorption entails different proteins having varying affinities for various solid surfaces. In these cases, a protein with a higher affinity for the surface adsorbs at a greater concentration than a protein with a lower affinity (*Lundqvist, 2013*).

The amino acid sequence forming the fundamental structure of proteins is one of the most crucial aspects of protein adsorption. Proteins are polypeptides [-NH-CHR-CO-] featuring functional groups and a main backbone structure that consists of the carboxyl terminus (C-terminus) and the amino terminus (N-terminus). The unique characteristics of the protein structure are derived from the functional group 'R'. Based on the chemical structure of their functional groups, proteins can be hydrophilic (polar), hydrophobic (nonpolar), or carry anionic/cationic charges. The functional group determines the active sites of proteins available for surface interaction, which influences protein adsorption on biomaterials (*Sanvictores & Farci, 2022*; *Stanciu & Diaz-Amaya, 2021*). Proteins generally fold in a way that exposes hydrophilic and charged groups to the external environment while positioning hydrophobic groups deep within the protein. During protein adsorption,

the adsorbed proteins spread out to expose their core, forming a monolayer of adsorbed proteins and releasing water molecules complexed with the native protein state (*Stanciu & Diaz-Amaya, 2021*).

Furthermore, the primary site for cell attachment in all proteins is the bioactive motif or domain Arg-Gly-Asp (RGD) (*Ryu et al., 2013*; *Wang et al., 2022*). Numerous ECM proteins, including collagen, laminin, vitronectin, fibronectin, fibrinogen, and others, have been found to contain the RGD sequence (*Love & Jones, 2013*; *Rowley et al., 2019*). Fibronectin, von Willebrand factor, and vitronectin proteins containing the RGD sequence interact with β1 integrin receptors on macrophages. Additionally, complement C3 fragments, fibrinogen, factor X, and high-molecular-weight kininogen bind to β2 integrins on macrophages to initiate initial monocyte attachment (*Kizhakkedathu & Conway, 2022*; *Piatnitskaia et al., 2024*; *Sheikh et al., 2015a*). The RGD sequence influences cell adhesion on biomaterials and the immunogenic cellular phenotype. Studies have shown that the RGD sequence of proteins affects the immunogenic cellular phenotype and function without altering the composition of the adsorbed protein layer (*Acharya et al., 2010*; *Acharya et al., 2011*; *Swartzlander et al., 2015*).

## Physicochemical property of biomaterial regulating protein adsorption and macrophage phenotype

The variation in protein adsorption according to the surface characteristics of bone grafts is briefly described in these sections. Additionally, this segment highlights the phenotypic differentiation of macrophages based on the physicochemical traits of bone grafts. Table 1 summarizes the pro-inflammatory and anti-inflammatory responses of macrophages in relation to the different physicochemical properties of bone grafts found in existing literature. Figure 3 illustrates macrophages' pro- and anti-inflammatory responses according to the surface attributes of bone grafts and their respective phenotypic differentiation markers.

### *Surface topography*

**Topography influencing protein adsorption.** The surface topography features, such as surface pores and porosity, impact protein adsorption on biomaterials. Variations in surface topography will result in differences in specific surface area (SSA) and surface charge density, which in turn affect the protein adsorption profile. An increase in porosity contributes to a greater specific surface area, facilitating the adsorption of high molecular-weight proteins (*Schlipf, Rankin & Knutson, 2013*; *Zhang et al., 2016*). For instance, a pore size of approximately six nm favors fibrinogen penetration, while a pore size of approximately two nm restricts fibrinogen penetration into the pores (*Zheng, Kapp & Boccaccini, 2019*; *Zhou & Hartmann, 2013*). A larger pore size, greater than 15 nm, enhances the adsorption of high molecular weight proteins, such as bone morphogenic protein (BMP) (*Kim et al., 2016*). However, pore sizes larger than the size of protein molecules have been reported to reduce protein activity (*Zhou & Hartmann, 2013*). **Topography influencing macrophage phenotypes.** The surface topography of biomaterials, particularly pore size, determines whether recruited macrophages adopt an M1 or M2 phenotype. Generally, larger porosities tend to favor the polarization of

**Table 1 Physicochemical properties of bone grafts influences macrophage phenotype.** The various physicochemical properties of bone grafts that affect the phenotypic regulation of macrophages are discussed. Additionally, the macrophage phenotypic markers, cell lines, or *in-vivo* models utilized are included as described in the literature.

| Physicochemical properties | Bone grafts | Cell lines/*In vivo* model | Macrophage phenotypes | Macrophage markers | References |
|---|---|---|---|---|---|
| **Surface topography** | | | | | |
| Rough surface | Titanium substrates | RAW 264.7 | M1 | IL-6, TNF-α | *Li et al. (2018)* |
| | Gold nanoparticles | BMDMs (Mouse) | M1 | IL-6, TNF-α, IL-1β | *Christo et al. (2016)* |
| Smooth surface | Mineralized collagen | THP-1 cells | M2 | IL-10 and IL-4 | *Li et al. (2020)* |
| | Titanium disks | Primary murine macrophages | M1 | IL-1β, IL-6 and TNF-α | *Hotchkiss et al. (2016)* |
| | Titanium disk | THP-1 | M2 | TGF-β, CCL18, MCR-1, CCL13, CD36 | *Zhang et al. (2019)* |
| Grooves & ridges | Poly-l-Lactic acid | RAW 264.7 | M2 | IL-1Ra, IL-10 | *Özcolak et al. (2024)* |
| Pits &bumps | Poly-l-Lactic acid | RAW 264.7 | M1 | IL-6, IL-1β | *Özcolak et al. (2024)* |
| Patterned | Polydimethylsiloxane | BMDMs (Mouse) | M2 | CD206, Arg-1, YM-1 | *McWhorter et al. (2013)* |
| **Surface wettability** | | | | | |
| Hydrophilicity | Titanium implants | C57BL/6 mice (10-week-old male) | M2 | IL-10, IL-4 | *Hotchkiss, Clark & Olivares-Navarrete (2018)* |
| | Modified SLA titanium discs | BMDMs | M2 | CD163, Arg1 | *Hamlet et al. (2019)* |
| | Titanium discs | RAW 264.7 | M2 | IL-10, TGF-β | *Gao et al. (2020)* |
| | Titanium implant | Sprague Dawley rats (8-week-old male) | M2 | Arg-1, IL-10 | *Ma et al. (2014)* |
| | Titanium implant | Human PBMCs | M2 | IL-1Ra, IL-4, IL-10, CCL-17, Arg-1 | *Abaricia et al. (2021b)* |
| Hydrophobicity | Titanium implants | C57BL/6 mice (10-week-old male) | M1 | IL1β, IL6 and TNFα | *Hotchkiss, Clark & Olivares-Navarrete (2018)* |
| | Silicon wafers | THP-1 | M1 | IL-1β, IL-6, TNF-α | *Visalakshan et al. (2019)* |
| | Titanium implants | C57BL/6 mice (10–12-week-old) | M1 | CD11b, CD68, CD86 | *Abaricia et al. (2021b)* |
| **Surface charge** | | | | | |
| Anionic charge | Mesoporous bioactive glass (MBG) | RAW 264.7; BMDMs | M2 | IL-10, Arg-1 | *Zeng et al. (2018)* |
| Cationic charge | Polyethyleneimine (PEI) | THP-1; RAW 264.7 | M1 | IL-12, TLR-4, TNF-α | *Mulens-Arias et al. (2015)* |
| | Co doped TiO2 | RAW 264.7 | M1 | TNF-α, IL-6, iNOS | *Li et al. (2019)* |
| | Titanium implant | Mouse J774.A1 macrophage | M2 | Arg-1, CD206, MR, CD163 | *Lee et al. (2016)* |
| **Surface porosity** | | | | | |
| Pore size | Collagen/chitosan (160 μm) | Male C57BL/6 J mice (6–8 weeks old) | M1 | CCR7, IL-1β and IL-6 | *Yin et al. (2020)* |
| | HA (4 μm) | RAW 264.7 | M1 | CD80, iNOS, TNF-α | *Yang et al. (2019)* |
| | Polydioxanone (34 μm) | BAT-GAL mice (7–9 months old) | M1 | iNOS and IL-1R1 | *Sussman et al. (2014)* |
| | PCL (40 μm) | Human peripheral blood Monocytes | M2 | IL-10, CD206, CD163 | *Tylek et al. (2020)* |
| | Collagen/chitosan (360 μm pore) | RAW 264.7 | M2 | TGF-β, IL-10, CD206 | *Yin et al. (2020)* |
| | HA (12 &36 μm) | RAW 264.7 | M2 | Arg-1, CD206, IL-10 | *Yang et al. (2019)* |

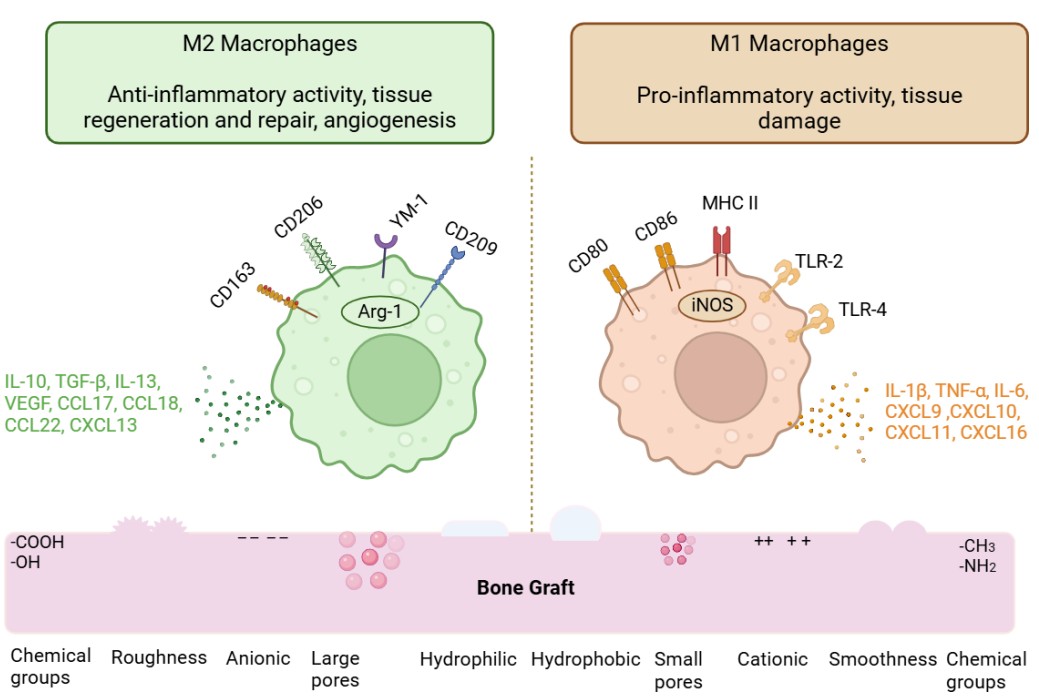

**Figure 3** **Physicochemical properties modulating macrophage phenotypes.** The macrophage phenotypes are influenced by the physicochemical characteristics of bone graft biomaterials. The M1 phenotype is promoted by surface features such as hydrophobicity, porosity, cationic charges, and methyl functional groups. In contrast, M2 macrophages are encouraged by hydrophilicity, increased surface roughness, anionic charges, and carboxyl functional groups. The various cytokines, chemokines, and surface markers of M1 and M2 macrophages are also discussed. (Image created using Biorender.com).

macrophages towards M2 phenotypes, while smaller pore sizes promote M1 phenotypes. For instance, collagen/chitosan scaffolds with a pore size of 360 µm encourage the polarization of M2 macrophages with pro-angiogenic and anti-inflammatory cytokine responses. Conversely, chitosan scaffold pores sized at 160 µm induce macrophages to exhibit a pro-inflammatory phenotype (*Yin et al., 2020*). Additionally, another study found that a PCL fiber scaffold with a pore size of 40 µm facilitated the differentiation of M2 macrophages (*Tylek et al., 2020*). *Yang et al. (2019)* study examined the modulation of macrophages by HA with pore sizes of four µm, 12 µm, and 36 µm. They found that the larger pore sizes of 12 µm and 36 µm promoted the M2 phenotype.

### Surface roughness

**Roughness influencing protein adsorption.** Enhanced protein adsorption occurs on rougher surfaces because they increase the biomaterial's surface area (*Lei et al., 2010*). For example, a titanium dioxide layer ($TiO_2$) with greater surface roughness created through chemical etching ($H_3PO_4/H_2O_2$ solution) enhances albumin adsorption compared to untreated $TiO_2$ substrates (*Pisarek et al., 2011*). Increased roughness alters the spatial arrangement of proteins and encourages protein conformational changes. Furthermore, as proteins are adsorbed onto the rough surface, the area occupied by denatured proteins will exceed that of proteins in their native state (*Niu et al., 2016*). For instance, rough silica

induces a conformational shift in adsorbed fibronectin compared to a flat surface, as the rough surface increases the area, prompting protein conformational changes (*Lei et al., 2010*).

**Roughness influencing macrophage phenotypes.** According to *Hotchkiss et al. (2016)*, a smooth titanium surface induced M1 polarization by expressing IL-1β, IL-6, and TNF-α, while a rougher titanium surface promoted the release of IL-4 and IL-10 from the M2 subpopulation of macrophages (*Hotchkiss et al., 2016*). Similarly, macrophages cultured on an 80 nm mechanically polished titanium dioxide (TiO$_2$) surface exhibited lower levels of inflammatory markers such as IL-1β, IL-6, TNF-α, MIP-1α, and MCP-1 compared to a surface topography with a 30 nm diameter (*Lü et al., 2015*).

Some research presents opposing views, suggesting that the smooth surface of osteogenic material enhances pro-inflammatory macrophages. According to an experimental design involving mineralized collagen with varying surface roughness, macrophages were polarized to M1 with high levels of inflammatory cytokines on a rough surface, including IL-6 and TNF-α. Conversely, the presence of a smooth surface led macrophages to express IL-10 (*Li et al., 2020*). Macrophages cultured on titanium surfaces with submicron-scaled surface roughness between 100 and 400 nm also indicated that as surface roughness increased, macrophages differentiated into M1 subtypes (*Li et al., 2018*).

### Surface chemistry (surface charge and functional groups)

**Surface chemistry influencing protein adsorption.** The electrostatic attraction between the protein and the biomaterial surface drives protein adsorption. Depending on the surface chemical composition of biomaterials, electrostatic interactions may either stimulate or inhibit protein adsorption. Specifically, the atoms on the surfaces of biomaterials and protein structures interact *via* charge-charge interactions, with opposing charges favoring protein adsorption (*Kyriakides, 2015*; *Zheng, Kapp & Boccaccini, 2019*). Another crucial factor to consider regarding charge and protein adsorption is the isoelectric point (pI) (*Moldoveanu & David, 2017*). The isoelectric point is defined as the pH of a solution at which the net charge of a protein is zero. When the pH of a solution is above a protein's pI, the protein is predominantly negatively charged; conversely, at a solution pH below the pI, the protein surface is predominantly positively charged (*Tokmakov, Kurotani & Sato, 2021*). For example, at a physiological pH of 7.4, bovine serum albumin (BSA) has an isoelectric point of 4.5 (indicating the pH of the solution is above the pI of albumin), while lysozyme has an isoelectric point of 11 (suggesting the pH of the solution is below the pI of lysozyme), indicating that BSA is negatively charged and lysozyme is positively charged (*Mueller, Zacharias & Rezwan, 2010*). Most serum proteins, like albumin and fibronectin, are negatively charged. An increase in negative surface charge contributes to reduced protein adsorption due to electrostatic repulsion, whereas a decrease promotes protein adsorption (*Zheng, Kapp & Boccaccini, 2019*).

Protein adsorption decreases on hydrophilic surfaces with functional groups such as hydroxyl (-OH) and carboxyl (-COOH) because the hydrogen bond between the surface and water is so strong that the protein cannot displace the interfacial water and get adsorbed on the surface of biomaterials. In contrast, hydrophobic surfaces containing amine (-NH2)

and methyl (-CH3) groups promote greater protein adsorption on their surfaces (*Rostam et al., 2015*; *Vogler, 2012*; *Zhou, Loppnow & Groth, 2015*).

**Surface chemistry influencing macrophage phenotypes.** On an anionic poly (acrylic acid) substrate, a decreased expression of IL-8 and increased IL-10 secretion from macrophages was observed. IL-10 and IL-1RA expression were found to be suppressed in response to the cationic functional groups of poly-dimethyl aminopropyl acrylamide (*Lee et al., 2019*). These results indicated that anionic surface charge promoted an anti-inflammatory or M2 modulation of macrophages, while cationic surface charge promoted a pro-inflammatory phenotype of macrophages. Another study by *Lee et al. (2016)* found that titanium implants modified with divalent cationic atoms like $Ca^{2+}$ and $Sr^{2+}$ enhanced the secretion of M2 markers such as Arginase 1 and mannose receptors while downregulating pro-inflammatory markers like TNF-α and IL-1β (*Lee et al., 2016*). Furthermore, macrophages released more TGF-β and less TNF-α and IL-6 when cultivated on magnesium calcium phosphate scaffolds (*Wang et al., 2016*). An inflammatory reaction is generally more likely to be triggered by cationic (positively charged) surfaces than by anionic (negatively charged) surfaces (*Li et al., 2021*).

The effects of surface chemistry on macrophage phenotype modification were examined using various self-assembling monolayers (SAMs) with distinct terminal groups, such as methyl (CH3), amine (NH2), hydroxyl (OH), and carboxyl (COOH) groups (*Zhou, Loppnow & Groth, 2015*). According to the study, the hydrophobic CH3 surface exhibited the strongest inflammatory response, leading to macrophage fusion and the production of inflammatory cytokines like TNF-α and IL-6. The least inflammation occurred on the hydrophilic COOH surface. Furthermore, in an *in-vivo* model involving BALB/c mice, the CH3 surface generated a thick fibrous capsule (*Zhou et al., 2016*).

### Surface wettability

**Surface wettability influencing protein adsorption.** Protein adsorption typically occurs more on hydrophobic surfaces than on hydrophilic ones, and proteins bond more strongly to hydrophilic surfaces. Greater protein adsorption happens on hydrophobic surfaces since fewer water molecules need to be displaced before protein adsorption (*Lin et al., 2011*). Additionally, certain serum proteins are attracted to hydrophilic surfaces. For example, the glycoprotein vitronectin, found in the ECM and plasma at concentrations of 200–400 µg/mL (*Mohamed et al., 2022*), is readily adsorbed onto hydrophilic surfaces (*McKiel, Woodhouse & Fitzpatrick, 2020*). Furthermore, adhesion-promoting proteins like fibrinogen and IgG2 favor hydrophilic surfaces, while adhesion-limiting proteins like albumin and fibronectin prefer hydrophobic surfaces (*Eslami-Kaliji et al., 2020*; *Wang et al., 2022*).

**Surface wettability influencing macrophage phenotypes.** A study comparing the macrophage polarization of sandblasted, large-grit, acid-etched (SLA) titanium and hydrophilic-modified SLA (modSLA) titanium revealed enhanced expression of M2 markers such as Arg1 and CD163 on the hydrophilic modSLA titanium surface. In contrast, the hydrophobic SLA titanium surface polarized macrophages towards M1 subsets and expressed inflammatory cytokines, including IL1β, IL6, and TNF-α (*Hamlet et al., 2019*).

On superhydrophilic TiO2 nanotubes, anti-inflammatory cytokines such as IL-10, TGF-β, and BMP-2 were overexpressed, while IL-6, TNF-α, and MCP-1 were downregulated. According to these studies, the hydrophilic titanium surface enhanced M2 phenotypes and promoted implant integration into the surrounding tissue by supporting osseointegration between the implant and bone (*Ma et al., 2014*; *Wang et al., 2018*). Hydrophobic surfaces stimulate inflammatory reactions in biomaterials by encouraging leukocyte adhesion, macrophage fusion, and the release of inflammatory cytokines. In contrast, hydrophilic surfaces promote anti-inflammatory properties by inhibiting leukocyte adhesion and macrophage fusion and decreasing the expression of pro-inflammatory cytokines (*Lv et al., 2018*; *Zhou & Groth, 2018*).

## Danger signals

Danger-associated molecular patterns (DAMPs) are molecules found in the intracellular space or hidden within the ECM that provoke an inflammatory response when released into the extracellular space (*McKiel, Woodhouse & Fitzpatrick, 2020*) during tissue injury (*Vénéreau, Ceriotti & Bianchi, 2015*). DAMPs include nuclear proteins such as HMGB1, heat shock proteins (HSPs), and elements of the extracellular matrix like fibronectin extra domain A (Fn EDA) and hyaluronan (*McKiel, Woodhouse & Fitzpatrick, 2020*; *McKiel & Fitzpatrick, 2018*). These danger-signaling molecules are typically absent in significant concentrations under physiological conditions. However, when they are present at substantial concentrations, they pose a danger to the microenvironment, leading to the activation of immune cells that work to eliminate the source of cellular distress or damage. These danger signals can activate pattern recognition receptors (PRRs) and toll-like receptors (TLRs) (*Ma, Jiang & Zhou, 2024*; *McKiel & Fitzpatrick, 2018*).

At the time of biomaterial implantation, the surrounding tissues are injured, and the damaged tissue releases DAMPs. These molecules bind to TLRs, initiating intracellular signaling events that lead to the production of pro-inflammatory cytokines. Upon binding to TLRs, they activate cytoplasmic adapter molecules that trigger cellular pathways such as nuclear factor kappa-light-chain-enhancer of activated B cells (NF-κB) and interferon regulatory factors associated with mitogen-activated protein kinase (MAPK) pathways. Activating these pathways produces inflammatory cytokines like TNF-α, IL-1, and IL-6 and chemokines through transcriptional and post-transcriptional mechanisms (*Tu et al., 2022*). The prolonged presence of DAMPs guides the wound toward a chronic inflammatory response, disrupting the balance between pro- and anti-inflammatory reactions during biomaterial implantation (*McKiel, Woodhouse & Fitzpatrick, 2020*).

HSPs are stress response proteins cells produce when exposed to chemical and physical stimuli. Examples of HSPs include HSP 70A, 70B, 60, 90, and 47. Biomaterial-associated cell stress or necrosis has been associated with the release of HSPs, which activate TLR 2 and 4, thereby inducing the inflammatory response at implantation sites (*Fang et al., 2011*; *Nonhoff et al., 2024*; *Wang et al., 2022*). When adsorbed on poly (methyl methacrylate) (PMMA) and polydimethylsiloxane, HSP 60 has been shown to activate NF-κB/AP-1-dependent SEAP (secreted alkaline phosphatase) and induce the expression of inflammatory cytokines (*McKiel & Fitzpatrick, 2018*).

FBR occurs in the absence of pathogens, and the inflammatory response is induced in the presence of DAMPs and an adsorbed protein layer. This type of immune activation that occurs in the absence of pathogens is referred to as sterile inflammation (*McKiel & Fitzpatrick, 2018*). The characteristics of sterile inflammation include the infiltration of neutrophils and macrophages and the expression of pro-inflammatory mediators such as IL-1β, TNF-α, and ROS (*Krysko et al., 2011*).

# BIOMATERIAL SURFACE MODIFICATION TO RESIST FBR

Various surface modification techniques have been explored to mitigate FBR. Over the past two decades, significant efforts have been directed toward developing advanced anti-fouling materials to reduce FBR's negative responses. Various approaches have been investigated to decrease FBR to biomaterials, focusing on the following strategies.

## Surface coating

Extensive research has focused on creating hydrophilic surfaces for biomaterials to minimize FBR. A key method involves modifying surfaces with zwitterionic polymers (*Shao & Jiang, 2015*; *Sun et al., 2014*). These polymers feature both anionic and cationic groups within a single molecular unit, leading to an overall neutral charge that effectively resists nonspecific protein adsorption in complex biological environments (*Blackman et al., 2019*; *Shao & Jiang, 2015*). This neutral charge facilitates the creation of a dense hydration layer on the biomaterial's surface, which inhibits protein adsorption through electrostatic repulsion. Notable examples of zwitterionic polymers, such as those containing carboxybetaine (CB) and sulfobetaine (SB) groups, demonstrate significant potential in reducing protein adsorption (*Blackman et al., 2019*).

Examples of zwitterionic polymers that effectively reduce protein adsorption on biomaterial surfaces include poly (2-hydroxyethyl methacrylate) (PHEMA) and poly (carboxybetaine methacrylate) (PCBMA) (*Zhou et al., 2024c*). Recently, a new class of zwitterionic polymers known as zwitterionic polypeptides (ZIPs) has been introduced (*Zhou et al. 2024a*). These hydrogels, which are characterized by alternating sequences of glutamic acid (E) and lysine (K), are specifically designed to minimize FBR. Zwitterionic polypeptides display anti-inflammatory properties and demonstrate strong resistance to FBR, enhancing the functional performance of implanted biomaterials (*Zhou et al. 2024a*; *Zhou et al., 2024c*).

Additionally, zwitterionic polymers, such as poly (sulfobetaine methacrylate) (PSB) (*Dong et al., 2021*) and poly (2-methacryloyloxyethyl phosphorylcholine) (PMPC) (*Park et al., 2014*), have been effectively utilized to create superhydrophilic antifouling coatings on hydrophobic substrates, leading to a significant reduction in FBR *in vivo*. Another promising antifouling biomaterial includes intrinsically disordered proteins (IDPs), derived from fused in sarcoma (FUS) proteins, which are rich in hydrophilic residues (*Chang et al., 2022*). When applied to biomaterial surfaces, these hydrophilic residues effectively prevent protein adsorption. The outstanding antifouling properties of zwitterionic polymers position them as highly promising candidates for minimizing foreign body reactions.

## Modification of material property

The physical properties of biomaterials have been extensively studied, highlighting their essential role in modulating the FBR to implanted materials. Intrinsic characteristics such as size, geometry, porosity, surface topography, and stiffness significantly influence cellular behavior and the overall FBR at both the molecular and cellular levels. For instance, nanoscale surface roughness has been shown to enhance protein adsorption, emphasizing the significance of surface topography in shaping biomaterial-host interactions (*Mariani et al., 2019*; *Zhou et al., 2024c*).

Material stiffness is a particularly influential property that governs macrophage adhesion and activation, both of which are central to FBR. Research on poly (ethylene glycol)-arginine-glycine-aspartic acid (PEG-RGD) hydrogels has demonstrated that softer hydrogels reduce macrophage activation, leading to a diminished FBR (*Scott, Kiick & Akins, 2021*). Further insights were provided by *Noskovicova, Hinz & Pakshir (2021)*, who investigated the effects of coating stiff silicone implants with a soft silicone layer. The study found that softer materials significantly reduced fibrosis formation and fibroblast activation, which are key contributors to developing a fibrous capsule around implants. These findings suggest that decreased material stiffness can lead to less fibrous encapsulation. However, this creates a challenge for applications such as bone regeneration, where biomaterials intended for load-bearing regions must maintain adequate mechanical strength to ensure structural stability and functional performance.

Surface topography is crucial in influencing the FBR, especially through its effect on surface properties. Studies have identified an optimal average surface roughness of about 4 $\mu$m for minimizing FBR in both *in vivo* models and human tissue samples (*Doloff et al., 2021*). This evidence underscores the importance of optimizing surface characteristics to decrease adverse immune responses and improve the biocompatibility of implanted materials.

## Incorporation of immunomodulatory agents

The reduction of FBR can be effectively achieved by incorporating immunomodulatory agents into the design of biomaterials. Applying functional groups to polymer coatings significantly decreases FBR. For example, a polymer derived from Z2 Y12, called poly (tetrahydropyran phenyl triazole) (PTHPT), has been used as a surface coating to combat FBR (*Zhou et al., 2024c*). *In vivo*, research shows that PTHPT coatings notably reduce capsule formation and fibrous tissue development in implants placed in the peritoneal cavity. Variants of the Z2-Y12 moiety, such as Met-Z2-Y12, have demonstrated greater effectiveness in reducing FBR responses, particularly as coatings for subcutaneous implants in mouse studies (*Wright et al., 2023*). Another promising category of immunosuppressive coatings includes phospholipids like phosphatidylcholine, phosphatidylethanolamine, sphingomyelin, and phosphatidylinositol. Research shows that applying phospholipids to biomaterial surfaces increases the transcription of anti-inflammatory genes in murine models (*Zhou et al., 2024c*).

## Biomimetic design

One alternative method to reduce FBR is using self-mimicking coatings on biomaterial surfaces, replicating the body's natural composition. Research indicates that certain intrinsic components can counteract FBR effectively. For example, albumin, the most prevalent plasma protein, has been thoroughly researched for its potential uses in this area (*Zhou et al., 2024b*). Applying albumin coatings to biomaterial surfaces has been proven to notably diminish macrophage adhesion and lessen the inflammatory response commonly observed after biomaterial implantation (*Hussain et al., 2020*; *Tao et al., 2020b*).

Another significant biomimetic strategy involves incorporating lipid bilayer structures that closely resemble the composition and functionality of cellular membranes. Polyphenol-based coatings, such as poly (tannic acid), have also been studied for their resemblance to red blood cell membranes at implant interfaces. These coatings demonstrate promising anti-biofouling properties and potential immunomodulatory effects on macrophages. Moreover, liposome coatings, which mimic natural cell membranes, offer an innovative approach to enhancing the biocompatibility of implanted biomaterials. Collectively, these self-mimicking strategies represent a promising direction for improving biomaterials' integration and functional performance while reducing adverse immune responses (*Tao et al., 2020a*; *Yang et al., 2021*).

## Drug-releasing surface coating

FBR to biomaterials can also be mitigated using surface coatings containing glucocorticoids, with dexamethasone being the most commonly used agent (*Khurana et al., 2014*). Recent studies have shown that the controlled release of glucocorticoids like dexamethasone (Dex) from biomaterial surfaces effectively diminishes FBR following implantation.

In addition to glucocorticoids, the controlled release of bioactive gas molecules has emerged as an innovative strategy to tackle FBR. For instance, surface coatings containing the tyrosine kinase inhibitor masitinib have been shown to significantly reduce collagen capsule thickness in mice 28 days post-subcutaneous implantation. Likewise, nitric oxide (NO), an essential signaling molecule, has been utilized to modulate fibroblast-mediated collagen deposition. It has been reported that the controlled release of NO from polymer surfaces decreases fibrotic capsule thickness (*Malone-Povolny et al., 2021*).

In addition to its antifibrotic effects, NO has been linked to enhanced angiogenesis and improved vascular stability. Studies have indicated a 77% increase in blood vessel formation one week after implantation in murine models. These findings highlight the potential of glucocorticoids and bioactive gas molecules as effective agents for reducing FBR and enhancing the integration and performance of implanted biomaterials (*Taylor et al., 2022*).

The strategies we previously explored for reducing FBR on biomaterial surfaces are promising in decreasing biomaterial rejection. Nonetheless, completely preventing non-specific protein adsorption on implant surfaces poses a notable challenge in practical scenarios. A deep understanding of the complex mechanisms involved in protein adsorption and subsequent immune cell interaction, along with their underlying molecular mechanism and pathways, is essential for successfully developing effective anti-FBR biomaterials.

## DISCUSSION

The primary goal of any bone graft biomaterial is to promote bone regeneration and integrate effectively with surrounding bone tissue. FBR remains a significant clinical challenge, affecting biomaterials' integration and their long-term effectiveness. This review highlights the significance of BAMPs in modulating FBR. It provides insights into how various components of BAMPs regulate macrophages. Macrophages are central to FBR's acute and chronic phases, influencing clinical outcomes such as bone graft integration or rejection.

The influence of adsorbed serum proteins (component of BAMPs) on the modulation of macrophage phenotypes is evident during the initial stages of FBR. Furthermore, research has explored serum proteins such as albumin, fibrinogen, vitronectin, and fibronectin that impact macrophage phenotypes, steering them toward either anti-inflammatory or pro-inflammatory subsets (*Eslami-Kaliji et al., 2023*; *Hussain et al., 2020*; *Zhou & Groth, 2018*; *Lee et al., 2019*; *McNally & Anderson, 2011*; *Sheikh et al., 2015a*; *Tao et al., 2020b*). The phenomenon governing protein adsorption and desorption processes on bone graft surfaces is termed as Vroman effect (*Wei et al., 2021*). These research findings highlight the importance of developing bone graft surface properties that encourage selective protein adsorption, which would steer macrophages towards anti-inflammatory subsets, enhancing integration and supporting bone regeneration.

The physicochemical characteristics (an element of BAMPs) of bone grafts, such as surface roughness, wettability, charge, and porosity, have been shown to influence macrophage phenotypes. Studies indicate that hydrophilic surfaces modified with zwitterionic coatings can enhance M2 polarization, fostering an anti-inflammatory environment conducive to graft integration and bone regeneration (*Dong et al., 2021*; *Lv et al., 2018*; *Park et al., 2014*). In contrast, hydrophobic surfaces tend to direct macrophages toward pro-inflammatory subsets and induce macrophage fusion and FBGC formation, thereby promoting fibrous capsule formation (*Zhou et al., 2016*; *Hamlet et al., 2019*). Additionally, surface charges significantly regulate macrophage subsets, with negatively charged surfaces promoting anti-inflammatory macrophages and positively charged surfaces favoring pro-inflammatory ones (*Li et al., 2021*). These findings also offer valuable insights into manufacturing bone grafts and for selecting suitable graft materials for clinicians with desired surface properties to reduce FBR and ensure integration with surrounding bone tissue.

When discussing the significance of physicochemical characterization of bone grafts and protein adsorption, one must not overlook the effect of danger signals (BAMPs component). The release of HMGB1 and HSPs from damaged cells during surgical procedures can activate TLRs on macrophages, increasing the likelihood of inflammation (*Tu et al., 2022*; *McKiel, Woodhouse & Fitzpatrick, 2020*). These findings highlight the importance of minimally invasive techniques in bone grafting procedures to reduce the release of danger signals from surgically damaged cells. Therefore, the design of bone graft surfaces must also consider preventing the adsorption of danger signaling molecules on biomaterials, which can guide macrophages to adopt anti-inflammatory subsets. This

approach will enhance the prospects of successful integration and the long-term success of bone grafts.

When discussing the factors influencing the FBR process for bone grafting, it is also essential to contemplate the critical considerations in selecting the most suitable graft bone biomaterials. Choosing the most suitable bone graft material for clinical applications requires a comprehensive evaluation of various factors that impact its effectiveness in tissue regeneration. These factors include the size, shape, and location of the bone defect, as well as the availability of donor tissue and the specific properties of the biomaterial (*Ebrahimi, 2017*; *Ferraz, 2023*). Specific properties include biological considerations, such as the integration timeline. Successful bone grafting procedure depends on the graft's ability to integrate with the surrounding bone tissue while demonstrating controlled biodegradation. An optimal biodegradation of bone grafts prevents the collapse of bone defects and promotes bone deposition and remodeling (*Sheikh et al., 2015b*). However, there is no evidence regarding the time required for bone graft integration and osseointegration, as these factors are influenced by the source of the material and the amount that is not completely degradable.

Considering the results of *in-vivo* studies on various bone grafts, autologous bone remains the gold standard for bone grafting procedures due to its exceptional osteogenic and osteoinductive properties, which have consistently demonstrated greater efficacy in tissue regeneration. While allografts offer some osteoinductive advantages, they also carry risks of immunogenic reactions and require rigorous screening. In contrast, xenografts and synthetic substitutes are osteoconductive but typically lack the complete regenerative capability of autografts. In load-bearing applications, such as dental and orthopedic implants, titanium and metallic grafts are often utilized due to their enhanced mechanical strength. Furthermore, xenografts have demonstrated promising clinical outcomes in oral and craniofacial defects (*Sallent et al., 2020*). Osteoconductive organic grafts such as chitosan and collagen promote bone matrix deposition and healing. However, they are better suited for non-load-bearing defects due to their limited mechanical properties (*Aibani et al., 2021*; *Signorini et al., 2023*).

Recent advancements in bone tissue engineering have consistently decreased the reliance on autografts while increasing the use of synthetic bone scaffolds (*Haugen et al., 2019*; *Sallent et al., 2020*). The popularity of synthetic bone grafts is attributed to their ease of handling, self-hardening properties, use of reproducible materials, and potential for large-scale production. Studies suggest that the effectiveness of bone regeneration with these grafts depends on factors like composition, size, shape, and particle porosity, which can be difficult to regulate when creating xenogeneic materials. Initial *in-vivo* investigations using animal models have yielded encouraging results for synthetic bone grafts; nevertheless, additional research with larger animals and human participants is essential to accurately assess their bone regeneration and integration abilities.

Recognizing the role of adsorbed serum-derived proteins associated with FBR is crucial, as they can serve as a double-edged sword in managing inflammation due to the function of serum proteins that may either enhance or suppress FBR. An exploratory study in this field can aid in developing bone graft surface properties that promote the adsorption of proteins that reduce inflammation while minimizing the adsorption of proteins that promote it.

Given that protein adsorption triggers FBR, a novel approach to identify the adsorbed proteome profile and potential conformational alterations of serum proteins could provide an effective solution for mitigating FBR.

## CONCLUSION

In summary, this review underscores the essential function of BAMPs in regulating the immune response to bone graft materials. Recent innovations in biomaterial surface modifications aimed at reducing FBR mark an exciting advancement in biomaterial research. While recent developments in surface alterations to minimize FBR by hindering protein adsorption are encouraging, it is important to note that completely preventing protein adsorption remains unattainable. This suggests that challenges related to FBR continue to exist. By integrating surface modification strategies with biomimetic surface properties that promote the adsorption of proteins that limit FBR, we could develop next-generation bone grafting materials with enhanced potential for bone regeneration and improved biological integration. As a result, forthcoming studies focusing on the proteomic profiling of adsorbed serum proteins (components of BAMPs) and identifying both FBR-promoting and FBR-limiting proteins may offer viable solutions to reduce the FBR response to bone grafts.

## ACKNOWLEDGEMENTS

The authors sincerely appreciate the support of the College of Graduate Studies at the University of Sharjah, UAE, which gave us access to database searches, seminars, and workshops that were beneficial in writing the review article. We also acknowledge the Research Institute for Medical and Health Sciences (RIMHS) for the research facilities it offers. We greatly appreciate BioRender's technical support in creating figures for the review.

### Funding

The authors received no funding for this work.

### Competing Interests

The authors declare there are no competing interests.

### Author Contributions

- Carel Brigi conceived and designed the experiments, performed the experiments, analyzed the data, prepared figures and/or tables, authored or reviewed drafts of the article, and approved the final draft.
- K.G. Aghila Rani conceived and designed the experiments, performed the experiments, analyzed the data, authored or reviewed drafts of the article, and approved the final draft.

- Balachandar Selvakumar conceived and designed the experiments, performed the experiments, analyzed the data, authored or reviewed drafts of the article, and approved the final draft.
- Mawieh Hamad conceived and designed the experiments, performed the experiments, analyzed the data, prepared figures and/or tables, authored or reviewed drafts of the article, and approved the final draft.
- Ensanya Ali Abou Neel performed the experiments, analyzed the data, prepared figures and/or tables, and approved the final draft.
- AR Samsudin conceived and designed the experiments, performed the experiments, analyzed the data, prepared figures and/or tables, authored or reviewed drafts of the article, and approved the final draft.

## Data Availability

This is a literature review.

## Supplemental Information

Supplemental information for this article can be found online at http://dx.doi.org/10.7717/peerj.19299#supplemental-information.

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

## FURTHER READING

**Amid R, Kheiri A, Kheiri L, Kadkhodazadeh M, Ekhlasmandkermani M. 2020.** Structural and chemical features of xenograft bone substitutes: a systematic review of *in vitro* studies. *Biotechnology and Applied Biochemistry* **68(6)**:1432–1452 DOI 10.1002/bab.2065.

**De Risi V, Clementini M, Vittorini G, Mannocci A, De Sanctis M. 2015.** Alveolar ridge preservation techniques: a systematic review and meta-analysis of histological and histomorphometrical data. *Clinical Oral Implants Research* **26(1)**:50–68 DOI 10.1111/CLR.12288.

**Deng F, Zhai W, Yin Y, Peng C, Ning C. 2021.** Advanced protein adsorption properties of a novel silicate-based bioceramic: a proteomic analysis. *Bioactive Materials* **6(1)**:208–218 DOI 10.1016/j.bioactmat.2020.08.011.

**Dwivedi R, Kumar S, Pandey R, Mahajan A, Nandana D, Katti DS, Mehrotra D. 2020.** Polycaprolactone as biomaterial for bone scaffolds: review of literature. *Journal of Oral Biology and Craniofacial Research* **10(1)**:381–388 DOI 10.1016/J.JOBCR.2019.10.003.

**Ge M, Ge K, Gao F, Yan W, Liu H, Xue L, Jin Y, Ma H, Zhang J. 2018.** Biomimetic mineralized strontium-doped hydroxyapatite on porous poly (l-lactic acid) scaffolds for bone defect repair. *International Journal of Nanomedicine* **13**:1707–1721 DOI 10.2147/IJN.S154605.

**Kaczmarek-Szczepańska B, Polkowska I, Małek M, Kluczyński J, Paździor-Czapula K, Wekwejt M, Michno A, Ronowska A, Pałubicka A, Nowicka B, Otrocka-Domagała I. 2023.** The characterization of collagen-based scaffolds modified with phenolic acids for tissue engineering application. *Scientific Reports* **13(1)**:1–12 DOI 10.1038/s41598-023-37161-6.

**Kalitheertha Thevar J-T, Nik Malek NAN, Abdul Kadir MR. 2019.** *In vitro* degradation of triple layered poly (lactic-co-glycolic acid) composite membrane composed of

nanoapatite and lauric acid for guided bone regeneration applications. *Materials Chemistry and Physics* **221**:501–514 DOI 10.1016/j.matchemphys.2018.09.060.

**Liang H-Y, Lee W-K, Hsu J-T, Shih J-Y, Ma T-L, Vo TTT, Lee C-W, Cheng M-T, Lee I-T. 2024.** Polycaprolactone in bone tissue engineering: a comprehensive review of innovations in scaffold fabrication and surface modifications. *Journal of Functional Biomaterials* **15(9)**:243 DOI 10.3390/jfb15090243.

**Lorenzi C, Leggeri A, Cammarota I, Carosi P, Mazzetti V, Arcuri C. 2024.** Hyaluronic acid in bone regeneration: systematic review and meta-analysis. *Dentistry Journal* **12(8)**:263 DOI 10.3390/DJ12080263.

**Oryan A, Alidadi S, Moshiri A, Maffulli N. 2014.** Bone regenerative medicine: classic options, novel strategies, and future directions. *Journal of Orthopaedic Surgery and Research* **9(1)**:18 DOI 10.1186/1749-799X-9-18.

**Pałka K, Pokrowiecki R. 2018.** Porous titanium implants: a review. *Advanced Engineering Materials* **20(5)**:1700648 DOI 10.1002/adem.201700648.

**Pina S, Rebelo R, Correlo VM, Oliveira JM, Reis RL. 2018.** Bioceramics for osteochondral tissue engineering and regeneration. *Advances in Experimental Medicine and Biology* **1058**:53–75 DOI 10.1007/978-3-319-76711-6_3.

**Sohn H-S, Oh J-K. 2019.** Review of bone graft and bone substitutes with an emphasis on fracture surgeries. *Biomaterials Research* **23(1)**:9 DOI 10.1186/s40824-019-0157-y.

**Titsinides S, Agrogiannis G, Karatzas T. 2019.** Bone grafting materials in dentoalveolar reconstruction: a comprehensive review. *Japanese Dental Science Review* **55(1)**:26–32 DOI 10.1016/j.jdsr.2018.09.003.

**Webber LP, Chan H-L, Wang H-L. 2021.** Will Zirconia implants replace titanium implants? *Applied Sciences* **11(15)**:6776 DOI 10.3390/app11156776.