# Peer review of "Decoding biomaterial-associated molecular patterns (BAMPs): influential players in bone graft-related foreign body reactions"

_PeerJ, doi:10.7717/peerj.19299_

## Round 0.1 · original submission · Major Revisions

Please address issues pointed by both reviewers and amend manuscript accordingly.

Reviewer 1 ·

Basic reporting

no comment

Experimental design

no comment

Validity of the findings

no comment

Additional comments

The manuscript by Carel Brigi et al., titled "Decoding Biomaterial Associated Molecular Patterns (BAMPs): Inûuential players in bone graft-related foreign body reactions", discussed the various aspects of FBR, BAMPs, its components and their role in initiation of FBR. And different types of bone grafting biomaterials and their physicochemical properties affecting protein adsorption and macrophage phenotype modulation were also discussed. The following suggestions should be considered:
1.In the abstract section of the paper, the author introduced the research background too much, and the innovation and main content of the paper are too simple, suggesting optimization.
2.The author should highlight the innovation of the paper in the introduction or abstract section, and the introduction section should also introduce whether there are relevant review analyses.
3.For the "1 Bone Grafting Biomaterials" section, composites based on bioceramics and polymers should be summarized. In this section, the authors are advised to add a table summarizing the advantages and disadvantages of various materials. For this section, 7.I recommend that the authors include the following references related to bioceramic/biopolymer composite materials for bone grafting: Advanced Functional Materials, 2023, 33:2214726, Bioactive Materials, 2021, 6:490-502, Journal of Advanced Research, 2022, 35:13-24, Advanced Science, 2018, 5(6):1700817.
4.For the "2 Bone grafting and the FBR" section, the subtitle is inappropriate because there is little introduction to bone grafting.
5.The analysis and discussion section of the paper is poor. The author not only lists the research results of others, but also needs to summarize and analyze these research results.

Reviewer 2 ·

Basic reporting

General Comment: The manuscript by Brigi et al., titled "Decoding Biomaterial Associated Molecular Patterns (BAMPs): Influential Players in Bone Graft-Related Foreign Body Reactions," aims to provide a comprehensive summary and classification of foreign body reactions (FBRs) associated with synthetic biomaterials employed in the treatment of fractured and dysfunctional bone tissues. However, to enhance the overall quality and scientific rigor of the manuscript, it is imperative to address the discrepancies and shortcomings highlighted in the detailed comments below:
Comment 1: The manuscript contains several typographical and grammatical errors that may hinder its clarity and professional presentation. The authors are advised to thoroughly review the entire manuscript for such errors. A detailed proofreading process or assistance from a professional language editing service is recommended to ensure linguistic accuracy and improve the overall readability and technical quality of the paper.

Experimental design

Comment 2: In the Introduction section, the authors aptly state: “Optimally, bone grafting biomaterial, be it metals, biopolymers, and composites, should not induce significant host inflammatory responses, should improve bone regeneration, and provide sustainable mechanical support (Daculsi et al., 2013; Zhao et al., 2021). That said, clinical experience with different biomaterials suggests that they precipitate undesirable host reactions or foreign body reactions (FBRs), which may render the biomaterial ineffective and functionless. Numerous literatures and case reports elaborating on failed bone grafts, due to FBR have been recorded (R. J. Adams, 2022; Badiee et al., 2022; Elakkiya et al., 2017; Kaing et al., 2011; Kamata et al., 2019; Nonhoff et al., 2024).”
To strengthen the manuscript, the authors are encouraged to emphasize, either in the conclusion or through an additional discussion section, critical considerations for selecting the most appropriate graft materials. This discussion should address strategies to mitigate late-stage adverse outcomes of foreign body reactions induced by implanted materials after primary surgical interventions. Highlighting such considerations would not only enhance the manuscript's practical relevance but also provide actionable insights for clinicians and researchers in the field.

Validity of the findings

Comment 3: This mini-review appears to resemble a section of a book chapter or thesis chapter rather than a standalone review article but can still be considered a mini-review. However, it lacks adequate emphasis on incorporating literature that highlights orthopedic cases, such as maxillofacial surgeries or critical-sized bone defects, where foreign body reactions (FBRs) have been reported with promising findings. This omission introduces potential selection biases, which may render the conclusions less evidence-based. To address this limitation, I recommend including a dedicated paragraph or at least some key phrases discussing studies conducted in various settings—in vitro, in vivo on animal models, in vivo on humans—and specifying the date intervals of these studies. Such additions would enhance the robustness, credibility, and evidence-based nature of the findings and conclusions presented, making the review more valuable to its audience.

Additional comments

Comment 4: The authors are encouraged to include a dedicated section that highlights strategies for mitigating Biomaterial-Associated Molecular Patterns (BAMPs) to minimize foreign body reactions (FBRs). This section should discuss the following approaches in detail such as Surface Coatings; Modification of Material Properties; Incorporation of Immunomodulatory Agents; Biomimetic Designs.

---

## Round 0.2 · accepted · Accept

All issues pointed out by the reviewers were adequately addressed, and the revised manuscript is acceptable in its current form.

Reviewer 1 ·

Basic reporting

no comment

Experimental design

no comment

Validity of the findings

no comment

Additional comments

no comment